# *Akkermansia muciniphila* Aspartic Protease Amuc_1434* Inhibits Human Colorectal Cancer LS174T Cell Viability via TRAIL-Mediated Apoptosis Pathway

**DOI:** 10.3390/ijms21093385

**Published:** 2020-05-11

**Authors:** Xin Meng, Jinrui Zhang, Hao Wu, Dahai Yu, Xuexun Fang

**Affiliations:** 1Key Laboratory for Molecular Enzymology and Engineering of Ministry of Education, College of Life Science, Jilin University, Changchun 130012, China; mengxin17@mails.jlu.edu.cn (X.M.); zjr13@mails.jlu.edu.cn (J.Z.); 2Vascular Biology Program, Department of Surgery, Boston Children’s Hospital and Harvard Medical School, Boston, MA 02115, USA; hao.wu3@childrens.harvard.edu

**Keywords:** Amuc_1434*, colorectal cancer cells, proliferation, apoptosis, TRAIL

## Abstract

Mucin2 (Muc2) is the main component of the intestinal mucosal layer and is highly expressed in mucous colorectal cancer. Previous studies conducted by our lab found that the recombinant protein Amuc_1434 (expressed in *Escherichia coli* prokaryote cell system, hereinafter termed Amuc_1434*), derived from *Akkermansia muciniphila*, can degrade Muc2. Thus, the main objective of this study was to explore the effects of Amuc_1434* on LS174T in colorectal cancer cells expressing Muc2. Results from this study demonstrated that Amuc_1434* inhibited the proliferation of LS174T cells, which was related to its ability to degrade Muc2. Amuc_1434* also blocked the G0/G1 phase of the cell cycle of LS174T cells and upregulated the expression of tumor protein 53 (p53), which is a cell cycle-related protein. In addition, Amuc_1434* promoted apoptosis of LS174T cells and increased mitochondrial ROS levels in LS174T cells. The mitochondrial membrane potential of LS174T cells was also downregulated by Amuc_1434*. Amuc_1434* can activate the death receptor pathway and mitochondrial pathway of apoptosis by upregulating tumor-necrosis-factor-related apoptosis-inducing ligand (TRAIL). In conclusion, our study was the first to demonstrate that the protein Amuc_1434* derived from *Akkermansia muciniphila* suppresses LS174T cell viability via TRAIL-mediated apoptosis pathway.

## 1. Introduction

The intestinal mucus layer is an important place for the interaction of intestinal cells and microbiota, which is essential for the maintenance of intestinal health and microecological balance [1]. Mucin 2 (Muc2) is the main component of the intestinal mucus layer, and it can be used as the sole source of nutrients for the growth of intestinal *Akkermansia muciniphila* [2]. *Akkermansia muciniphila* can use the sugar chain of Muc2 as a carbon source, and degrades it via the *Akkermansia-muciniphila*-encoded glycosidase [3]. We found that the protease encoded by *Akkermansia muciniphila*, Amuc_1434*, is the only currently reported protease enzyme able to degrade the Muc2 core protein [4].

Muc2 is involved in the development of colorectal cancer (CRC), which is a major health problem in the world, with more than 1.2 million patients diagnosed with colorectal cancer each year and nearly 600,000 deaths [5,6,7]. Compared with the normal mucosa, the expression of Muc2 in CRC tissues was significantly increased [8]. Therefore, restoration of a normal expression level of Muc2 may control the development of CRC.

The expression of Muc2 is regulated by tumor protein 53 p53 protein transcriptional regulation in some cell lines, and the immune reactivity of Muc2 is negatively correlated with changes of p53 [9,10]. For example, the expression level of p53 in mucous colorectal cancer is low, while the expression level of Muc2 is relatively high [11]. Mutation of p53, which is a tumor-suppressor gene, is responsible for 70% of CRC cases, so failure of apoptosis is a major factor in the transition from adenoma to CRC [12,13]. p53 also participates in the tumor-necrosis-factor-related apoptosis-inducing ligand (TRAIL)-mediated apoptotic pathway, which plays a role in the apoptotic process of colorectal cancer cells [14,15]. As a result, activation of the apoptosis of colon cells is of vital importance for controlling the progression of CRC [16].

Tumor-necrosis-factor-related apoptosis-inducing ligand (TRAIL) is a member of the TNF superfamily, which also induces malignant cell apoptosis in vitro and in preclinical cancer models [17,18,19,20]. Gene mutations in the TRAIL pathway are related to the occurrence of human tumors [21,22,23,24], but TRAIL has little or no toxicity towards normal cells [19]. TRAIL is expressed in a variety of tissues, such as small intestine, colon, spleen, thymus, prostate, and placenta [25,26,27]. TRAIL binds to DR4 and DR5, which are the death receptors, and recruits Fas-associated death domain (FADD) proteins by interacting with the death domain. FADD binds to the death effector domain on cysteinyl aspartate specific proteinase 8 (caspase 8) to form the death-induced signal complex (DISC), which is responsible for triggering the apoptotic cascade reaction [28,29,30,31]. TRAIL can also activate mitochondrial apoptosis pathways via activated-caspase-8-mediated B-cell lymphoma-2 interacting-domain death agonist (Bid) lysis, leading to cell apoptosis [32]. CRC is resistant to TRAIL-driven apoptosis when the TRAIL signaling mechanism is defective (e.g., when TRAIL receptor is downregulated and anti-apoptotic protein level is increased) [18]. However, studies have shown that by inducing the expression of death receptor 4 (DR4) and death receptor 5 (DR5) and upregulating TRAIL levels, the sensitivity of CRC cells to TRAIL-induced apoptosis can be enhanced [33,34,35]. Additionally, the generation of excessive reactive oxygen species (ROS) and destruction of the integrity of the mitochondrial membrane play an important role in inducing apoptosis [36,37].

Studies have shown that *Akkermansia muciniphila* is an intestinal symbiont colonizing the mucosal layer that can degrade human Muc2 and pig mucin5AC (Muc5AC) through its own proteolytic enzyme [38,39,40]. Since Muc2 is related to the progress of CRC, we investigated the effect of Amuc_1434* on the proliferation, cycle, and apoptosis of LS174T colorectal cancer cells expressing Muc2 and the possible mechanism of action in the current study. It was found that Amuc_1434* inhibited the proliferation of LS174T cells, interfered with the normal cell cycle of LS174T, and promoted cell apoptosis, and also induced intracellular ROS production and damaged mitochondrial membrane potential. In addition, Amuc_1434* activated both the exogenous death receptor and endogenous mitochondrial pathway by upregulating TRAIL. To sum up, Amuc_1434* inhibits LS174T cell viability via the TRAIL-mediated apoptosis pathway.

## 2. Results

### 2.1. Amuc_1434* Inhibited the Proliferation of LS174T Cells

The effect of Amuc_1434* on the growth of LS174T cells was detected by 3-(4, 5-dimethylthiazol-2-yl)-2,5-diphenyltetrazolium bromide (MTT) assay. In this experiment, Human epidermal melanoma (HEM) cells that did not express Muc2 were selected as controls. Amuc_1434* inhibited the proliferation of LS174T cells in a concentration-dependent manner at concentrations ≥ 8 μg/mL, and the cell survival rate was 70% when LS174T cells were treated with Amuc_1434* at a concentration of 64 μg/mL (Figure 1A). However, more than 90% cell viability was observed after HEM cells were incubated with 64 μg/mL Amuc_1434* (Figure 1B). This indicated that Amuc_1434* had no cytotoxicity to HEM cells. In addition, Muc2 was not expressed by HEM cells, which indicated that the inhibitory effect of Amuc_1434* on LS174T cell proliferation may be related to its ability to degrade Muc2.

### 2.2. Effects of Amuc_1434* on Cell Cycle of LS174T

A dose of 8 μg/mL Amuc_1434* inhibited the proliferation of LS174T cells, while 64 μg/mL Amuc_1434* also had an inhibitory effect on the proliferation of LS174T cells. Hence, these two concentrations were used for the subsequent experiments. The rate of cell proliferation is influenced by the regulation of cell cycle [41]. Once cell proliferation is affected, it often manifests as a change in the composition of the cell cycle [42]. Therefore, flow cytometry was used to detect the effect of Amuc_1434* on the cell cycle of LS174T cells.

The G0/G1 phase accounted for 52.97% of the total cell cycle in the control group, 57.37% of the total cell cycle in the low-concentration group, and 63.53% of the total cell cycle in the high-concentration group (Figure 2A). Therefore, Amuc_1434* induced G0/G1-phase cell-cycle arrest in LS174T cells. In addition, an effect of Amuc_1434* was observed on the expression of p53, which is the tumor suppressor controlling the initiation of the cell cycle. Compared with the control, the expression of p53 protein was upregulated by Amuc_1434* in a dose-dependent manner when compared with the control (Figure 2B). Thus, these results indicated that Amuc_1434* interferes with the LS174T cell cycle.

### 2.3. Amuc_1434* Induced Apoptosis of LS174T Cells

There is a dynamic balanced relationship between cell proliferation and apoptosis regulated by multiple genes under normal circumstances. Any abnormality in one of the links breaks this balance and causes excessive cell proliferation, which eventually leads to tumorigenesis [43]. Apoptosis (the opposite of cell proliferation) plays an important role in the inhibition of cancer cell proliferation. Flow cytometry was used in the current study to detect the apoptosis of LS174T cells, which were treated for 24 h with 8 μg/mL and 64 μg/mL Amuc_1434*. The apoptosis rate was 9.73% in the control group, 19.32% in the low-concentration (8 μg/mL) treatment group, and 25.15% in the high-concentration (64 μg/mL) treatment group (Figure 3). Thus, the apoptosis of LS174T cells was promoted by Amuc_1434* in a concentration-dependent manner.

### 2.4. Amuc_1434* Promoted the Change of Cellular Redox Status and Mitochondrial Dysfunction in LS174T Cells

In the early stages of apoptosis, the level of intracellular reactive oxygen species (ROS) increases [44]. In this study, LS174T cells treated with Amuc_1434* were co-incubated with the probe 2,7-Dichlorodi-hydrofluorescein diacetate (DCFH-DA) for flow cytometry detection. When compared with the control group, the ROS levels in LS174T cells increased by 18.45% after treatment with Amuc_1434* at 8 μg/mL, and increased more significantly (67.84%) after treatment with Amuc_1434* at 64 μg/mL (Figure 4A). Thus, Amuc_1434* increased the content of ROS in LS174T cells. Excessive production of ROS increases oxidative stress, which can lead to mitochondrial damage in human cells [45]. Hence, we investigated the effect of Amuc_1434* on the mitochondrial membrane potential of LS174T cells. Compared with the control group, JC-1 mainly existed as a monomer after treatment with Amuc_1434* at 8 μg/mL and 64 μg/mL (Figure 4B). This suggested that Amuc_1434* decreased the mitochondrial membrane potential of LS174T cells. Thus, Amuc_1434* may indirectly promote mitochondrial dysfunction by increasing the generation of intracellular ROS.

### 2.5. Amuc_1434* Activated the LS174T Cells’ Apoptotic Pathway via TRAIL

The exogenous death receptor pathway and the endogenous mitochondrial pathway are the two main pathways of apoptosis [46]. Each pathway is regulated via multiple proteins. Both the pathways are connected and influenced by each other [46]. Tumor necrosis factor-related apoptosis-inducing ligand (TRAIL, a member of the TNF family) and the caspase family play an important role in the initiation and maintenance of apoptosis in the exogenous death receptor pathway [47]. We examined the expression of apoptosis-related proteins in the exogenous death receptor pathway to investigate the mechanism by which exogenous Amuc_1434* induced apoptosis in LS174T cells. Compared with the control group, Amuc_1434* upregulated the apoptosis-inducing ligand TRAIL, the death receptor 4 and the death receptor 5 (DR4 and DR5) based on the ligand TRAIL (Figure 5A). Statistical analysis demonstrated that there was a 2-fold increase in the expression of DR4 and DR5 by 64 μg/mL Amuc_1434* when compared to the control group. The expression levels of cleaved- cysteinyl aspartate specific proteinase 8 (caspase 8) and cleaved- cysteinyl aspartate specific proteinase 3 (caspase 3) were also upregulated after treatment with low and high concentrations of Amuc_1434*. The activation ratio of caspase 8 and caspase 3 after treatment with 64 μg/mL Amuc_1434* was more than 2-fold higher than the control group.

Caspase 8 not only participates in the exogenous death receptor pathway, but also takes part in the mitochondrial endogenous pathway [32,48]. Therefore, we next investigated the effect of Amuc_1434* on the expression of apoptosis-related proteins in the mitochondrial pathway. Both 8 μg/mL and 64 μg/mL of Amuc_1434* upregulated the expression of mitochondria-associated pro-apoptotic factors B-cell lymphoma-2 interacting-domain death agonist (Bid), B-cell lymphoma-2-Associated X Protein (Bax), Cytochrome c (CytC), and Endonuclease G (EndoG), and downregulated the expression of mitochondria-related anti-apoptotic factor Bcl-2 when compared with the control group (Figure 5B). The above results indicated that Amuc_1434* activated the death receptor and mitochondrial pathways of apoptosis.

## 3. Discussion

Mucins play a role in carcinoma cell and tumor biology by enhancing cancer cell growth, proliferation, survival, and immune evasion [49]. Mucins are thought to provide cancer cells with a physicochemical barrier and a protective shield against external damage [50,51]. It has been reported that overexpression of Muc2 in CRC helps tumor cells to evade recognition by anti-tumor immune effector molecules, which contributes to the development of CRC [52]. Hence, mucin depletion could deprive tumor cells of a critical protective framework.

*Akkermansia muciniphila* is a key component of the gut microbiome with many biological functions, which can regulate the level of Muc2 through its own proteolytic enzyme [53,54,55]. Previous studies done by our group demonstrated that that Amuc_1434* has the ability to degrade Muc2 and is mainly expressed in the colon in mice. In the current study, we investigated the effect of Amuc_1434* on the growth of Muc2-expressing colorectal cancer LS174T cells. MTT assay found that Amuc_1434* inhibited the proliferation of LS174T cells at concentrations ≥ 8 μg/mL (Figure 1A). Moreover, Amuc_1434* was not toxic against the non-Muc2-expressing HEM cells (Figure 1B). This suggested that the inhibition of Amuc_1434* on the proliferation of LS174T cells might be related to the degradation of Muc2.

Reduced growth can be attributed to cell-cycle defects [56]. It was found that the proportion of G0/G1 phase in the whole cell cycle increased by 4.4% and 10.36% after treatment of LS174T cells with Amuc_1434* at 8 μg/mL and 64 μg/mL when compared with the respective control groups (Figure 2A). Amuc_1434* induced G0/G1-phase arrest of the LS174T cell cycle in a concentration-dependent manner. In addition, the orderly running of the cell cycle requires strict regulation of the genes involved. p53 is a tumor suppressor which controls the initiation of the cell cycle. Western blot analysis demonstrated that the expression of p53 in LS174T cells was increased after treatment with 8 μg/mL and 64 μg/mL Amuc_1434* in a concentration-dependent manner (Figure 2B). This indicated that Amuc_1434* might block the normal cell cycle of LS174T cells and inhibit the proliferation of LS174T cells by upregulating the expression of p53. Mutation of p53 is seen in a majority of CRC cases. The results of our study indicated that Amuc_1434* may play a major role in the management of CRC.

Previous studies have shown that the level of apoptosis induced in tumor cells is directly related to the inhibition of tumor cell proliferation [57,58,59]. ROS also play an important role in the process of apoptosis [44]. Using flow cytometry, we demonstrated in the current study that Amuc_1434* promoted the apoptosis of LS174T cells and increased the ROS content in LS174T cells (Figure 3 and Figure 4A). High levels of ROS damage mitochondria [60]. Fluorescence microscopy showed that Amuc_1434* decreased the mitochondrial membrane potential of LS174T cells (Figure 4B). Studies have shown that TRAIL plays an important role in inducing apoptosis of CRC cells [61]. In the exogenous apoptotic pathway, TRAIL binds to the receptors DR4 and DR5 to trigger the caspase cascade reaction to induce cell apoptosis. TRAIL additionally induces apoptosis via induction of the mitochondrial pathway and damaging the mitochondrial inner membrane [61]. Results obtained from Western blotting revealed that Amuc_1434* upregulated the expression of death ligand TRAIL and its receptors (DR5 and DR4) (Figure 5A). It also induced the activation of caspase 8 and caspase 3 (Figure 5A). This suggested that Amuc_1434* may activated the whole exogenous death receptor pathway by upregulating the expression of TRAIL to mediate cell apoptosis. In addition, it was also found that Amuc_1434* decreased/increased the expression of apoptosis-related proteins in the mitochondrial pathway of LS174T cells (Figure 5B). Amuc_1434* (8 μg/mL and 64 μg/mL) significantly decreased the expression of anti-apoptotic protein Bcl-2 and increased the expression of pro-apoptotic proteins including Bid, Bax, EndoG, and CytC (Figure 5B). A plausible explanation for these findings is that caspase 8, which is activated by Amuc_1434*, participates in the mitochondrial pathway, causing changes in the expression of apoptosis-related proteins in the mitochondrial pathway. Among these, EndoG is associated with DNA damage [62]. Thus, the upregulation of EndoG protein could be one of the factors responsible for increasing the expression of p53. The anti-apoptotic factor Bcl-2 also has an inhibitory effect on the expression of p53 [63]. Thus, the downregulation of Bcl-2 protein may also be a reason for the increased p53 expression. Additionally, p53 negatively regulates Muc2 and participates in the TRAIL mediated apoptotic pathway [9,10,11,12,13,14,15], so the upregulation of TRAIL by Amuc_1434* may be a reason for its degradation of Muc2. In addition, studies have shown that there is negative regulation between Bcl-2 protein and intracellular ROS [32]. Therefore, the downregulation of the expression of Bcl-2 by Amuc_1434* may have been the reason for the increase of intracellular ROS. Taken together, these data indicated that Amuc_1434* suppressed LS174T cell viability via the TRAIL-mediated apoptosis pathway.

## 4. Materials and Methods

### 4.1. Materials

Amuc_1434* protein was produced according to our previous study [4]. BCA (bicinchonininc acid) Protein Assay Kit (C503051) was purchased from Sangon Biotech Co. Ltd. (Shanghai, China). Dulbecco’s modified Eagle’s medium (MA0212, DMEM, including 4.5 g/L d-glucose, 584 mg/L L-glutamine and 110 mg/L sodium pyruvate) and fetal bovine serum (A31607) for LS174T and HEM cell cultures were purchased from Dalian Meilun Biotechnology Co. LTD. (Dalian, China) and Life Technologies (Scotland, UK), respectively. MTT (MB4698) was purchased from Aladdin (Shanghai, China). Cell cycle kit (BB-4104-2) and Annexin V-FITC Apoptosis Detection kit (BB-4101-2) were purchased from BestBio Science (Shanghai, China).

Reactive oxygen species assay kit (S0033) and mitochondrial membrane potential assay kit with JC-1 (C2006) were purchased from Beyotime Biotechnology (Shanghai, China). Primary antibodies included anti-β-actin mouse pAb (T0022), anti-GAPDH mouse pAb (T0004), anti-p53 rabbit pAb (AF0879), anti-TNFSF10 rabbit pAb (DF6861), anti-caspase 8 rabbit pAb (AF6442), anti-cleaved-caspase 8 rabbit pAb (AF5267), anti-anti-caspase 3 rabbit pAb (AF6311), anti-cleaved-caspase 3 rabbit pAb (AF7022), anti-DR4 rabbit pAb (AF0304), anti-DR5 rabbit pAb (DF6368), anti-Bid rabbit pAb (DF6016), anti-Bcl2 rabbit pAb (AF6139), anti-Bax rabbit pAb (AF0120), anti-CytC rabbit pAb (AF0146), anti-EndoG rabbit pAb (DF8541) (Affinity Biosciences, Queensland, Australia). Secondary antibodies included horseradish peroxidase (HRP)-conjugated goat anti-rabbit antibody (S0001) and HRP-conjugated goat anti-mouse antibody (S0002) (Affinity Biosciences, Queensland, Australia).

### 4.2. Cell MTT Assay

Cell viability was assessed through 3-(4,5-dimethylthiazol-2-yl)-2,5-diphenyltetrazolium bromide (MTT) assays. LS174T and HEM cells were seeded in 96 well plates (5 × 10^3^ cells/well) in DMEM (10% FBS) and cultured overnight. The cells were treated with different concentrations (0, 2, 8, 32, and 64 μg/mL) of Amuc_1434* in DMEM for 24 h. Subsequently, 20 μL of 5 mg/mL MTT reagent (MB4698, Aladdin, Shanghai, China) was added to each well for 4 h at 37 °C. Next, 150 μL of dimethyl sulfoxide (DMSO, DH105-2, BEIJING DINGGUO CHANGSHENG BIOTECHNOLOGY CO. LTD., Beijing, China) was added to each well to dissolve the formazan crystals. The absorbance of each well was measured at 570 nm wavelength. The cell viability was calculated for each well as A570 treated cells/A570 control cells × 100%.

### 4.3. Flow Cytometry Analysis

LS174T cells were seeded overnight in 6 well plates (10^5^ cells/well) for cell-cycle analysis. The cells were treated with various concentrations (0, 8, and 64 μg/mL) of Amuc_1434* for 24 h. Subsequently, the cells were washed with phosphate-buffered saline (PBS, 8 mM Na_2_HPO_4_, 136 mM NaCl, 2 mM KH_2_PO_4_, 2.6 mM KCl) and fixed in pre-cooled ethanol at −20 °C for 1 h. The cells were washed and centrifuged twice at 1000× *g* for 10 min. Collected cells were resuspended in 500 μL PBS with 20 μL RNase A from Cell cycle kit (BB-4104-2, BestBio Science, Shanghai, China) for 30 min at 37 °C. The cells were filtered with a cell strainer (aperture: 0.0374 mm), and then 400 μL propidium iodide (PI) was added from the Cell cycle kit (BB-4104-2, BestBio Science, Shanghai, China) for 30 min at 4 °C. The cells were then analyzed by flow cytometry (BD Bioscience, NY, USA).

LS174T cell suspension was adjusted to a density of 10^5^ mL^−1^, and the cells were seeded in 6 well plates overnight for analysis of cell apoptosis. LS174T cells were then incubated with different concentrations (0, 8, and 64 μg/mL) of Amuc_1434* in DMEM for 24 h. After 24 h, the percentage of apoptotic cells was determined using an Annexin V-FITC Apoptosis Detection Kit (BB-4101-2, BestBio Science, Shanghai, China). The cells were collected and added to 400 µL of 1× Annexin V Binding Buffer, 5 μL of annexin V-FITC and 5 μL of PI. Flow cytometry (BD Bioscience, NY, USA) was used to analyze apoptosis.

LS174T cells (10^5^ cells/well) were incubated with varying concentrations (0, 8, and 64 μg/mL) of Amuc_1434* in DMEM for 24 h to analyze the levels of ROS. Cells were harvested and washed with PBS and suspended in 10 μM 2,7-dichlorodihydrofluorescein diacetate (DCFH-DA, Beyotime Biotechnology, Shanghai, China) for 30 min at 37 °C. The cells were then washed twice with DMEM and subjected to flow cytometry (BD Bioscience, NY, USA) analysis.

### 4.4. JC-1 Staining and Mitochondrial Transmembrane Potential ΔΨm (MMP) Measurement

LS174T cells were incubated with different concentrations (0, 8, and 64 μg/mL) of Amuc_1434* in DMEM for 24 h, washed twice in PBS, and treated with 5 μg/mL lipophilic cationic probe JC-1 (Beyotime Biotechnology, Shanghai, China) for 20 min at 37 °C. JC-1 monomers were detected at excitation and emission wavelengths of 514–529 nm and JC-1 aggregates were detected at 585–590 nm. Differences in the color intensity ratio indicated the levels of ΔΨm.

### 4.5. Western Blot

LS174T cells were treated with different concentrations of Amuc_1434*. Total proteins were separately extracted from LS174T cells and the total protein concentration was measured using a BCA Protein Assay Kit (C503051, Sangon Biotech Co. Ltd., Shanghai, China). The same amount of protein was loaded into each well for 12% sodium dodecyl sulfate-polyacrylamide gel electrophoresis (SDS-PAGE) and transferred to polyvinylidene fluoride (PVDF) membranes (Millipore, CA, USA). The PVDF membranes were blocked with 10% nonfat dry milk (Solarbio, Beijing, China) in TBST (10 mM Tris-HCl, 150 mM NaCl and 2% Tween-20) at room temperature for 3 h, and incubated with primary antibodies (dilution ratio 1:1000, using PBS) at 4 °C overnight. Subsequently, PVDF membranes were washed three times with TBST for 15 min and incubated with HRP-conjugated secondary antibodies for 1 h. Subsequently, the PVDF membranes were washed three times for 15 min with TBST. The protein was detected with a super-sensitive electrochemiluminescence (ECL) luminescence reagent (MA0186, Meilunbio, Dalian, China).

### 4.6. Statistical Analysis

Each experiment was repeated three times. Graphpad Prism 7.0 (GraphPadSoftware, La Jolla California, USA) was used for statistical analysis. Protein expression levels and JC-1 staining were quantified using Image J (National Institutes of Health, Bethesda, Maryland, USA). Statistical analysis expression of the results is as mean ± standard deviation and presentations in graphics are mean ± S.E.M. Univariate analysis of variance was used for group comparison. * *p*-value < 0.05, ** *p*-value < 0.01. *p*-value < 0.05 was considered statistically significant.

## 5. Conclusions

In conclusion, the present study suggested that Amuc_1434* inhibited the proliferation and cell cycle of LS174T cells, promoted the apoptosis of LS174T cells, upregulated the ROS level in LS174T cells, and caused mitochondrial dysfunction of LS174T cells. We also demonstrated that Amuc_1434* activates the death receptor pathway and mitochondrial pathway of apoptosis, which may play a critical role in Amuc_1434*-induced inhibition of colorectal cancer cell viability (Figure 6). The results also provided evidence that Amuc_1434* may have a role in controlling CRC, and further studies are required to gain an in-depth understanding of the function of protease Amuc_1434* in the management of CRC.

## Figures and Tables

**Figure 1 ijms-21-03385-f001:**
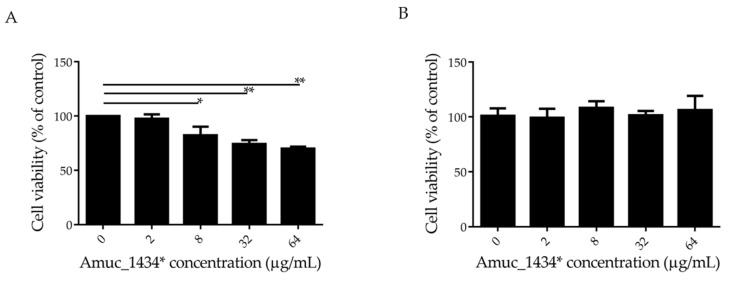
Amuc_1434* inhibited the proliferation of colorectal cancer LS174T cells. (**A**) LS174T cells and (**B**) Human epidermal melanoma (HEM) cells treated with various concentrations (0, 2, 8, 32, and 64 μg/mL) of Amuc_1434* for 24 h. The viability of LS174T and HEM cells was detected via 3-(4, 5-dimethylthiazol-2-yl)-2,5-diphenyltetrazolium bromide (MTT) assay. (Data are expressed as mean ± standard deviation, *n* = 3, *: *p* < 0.05; **: *p* < 0.01.).

**Figure 2 ijms-21-03385-f002:**
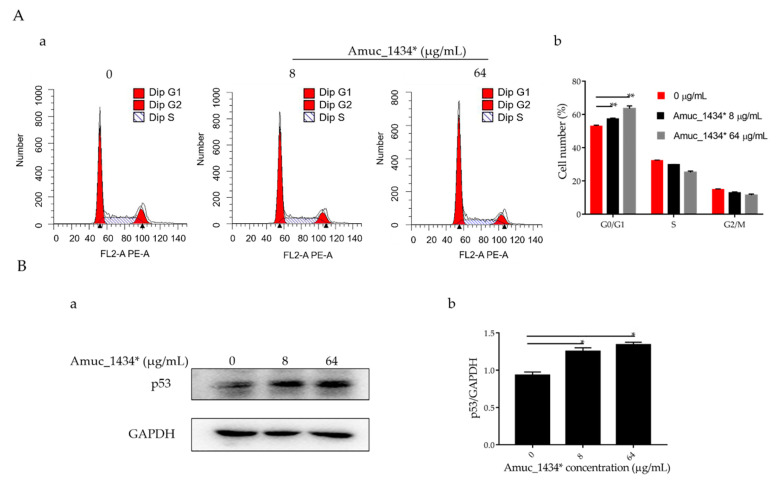
Amuc_1434* treatment induced G0/G1-phase cell-cycle arrest. (**A**) Cell cycle analysis. (a) LS174T cells were treated with Amuc_1434*, and cell-cycle distribution was evaluated by flow cytometry. (b) Percentages of G0/G1 phase of the cell cycle in LS174T cells are shown. (Data are expressed as mean ± standard deviation, *n* = 3, *: *p* < 0.05.) (**B**) Western blotting analysis for the expression level in LS174T cells of tumor protein 53 (p53) (a), which controls the start of the cell cycle, after treatment with Amuc_1434*. GAPDH was used as the loading control. (b) Quantification of p53 expression levels in LS174T cells. (Data are expressed as mean ± standard deviation, *n* = 3, *: *p* < 0.05.).

**Figure 3 ijms-21-03385-f003:**
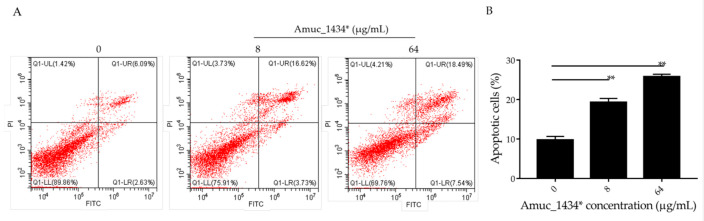
Amuc_1434* promoted cell apoptosis of colorectal cancer LS174T cells. (**A**) Relative cell apoptosis rate was detected via flow cytometric analysis with annexin V-FITC and PI double staining. LS174T cells are shown in the scatter plots, in which the upper left quadrant identifies necrotic cells (FITC−/PI+), the upper right quadrant identifies late-apoptotic cells (FITC+/PI+), the lower left quadrant identifies live cells (FITC−/PI−), and the lower right quadrant identifies early-apoptotic cells (FITC+/PI−). The apoptosis rate = percentage of cells in early apoptosis + percentage of cells in late apoptosis. (**B**) Percentages of apoptotic cells are indicated in a representative histogram; the data are expressed as mean ± standard deviation, *n* = 3, **: *p* < 0.01 compared to untreated cell group.

**Figure 4 ijms-21-03385-f004:**
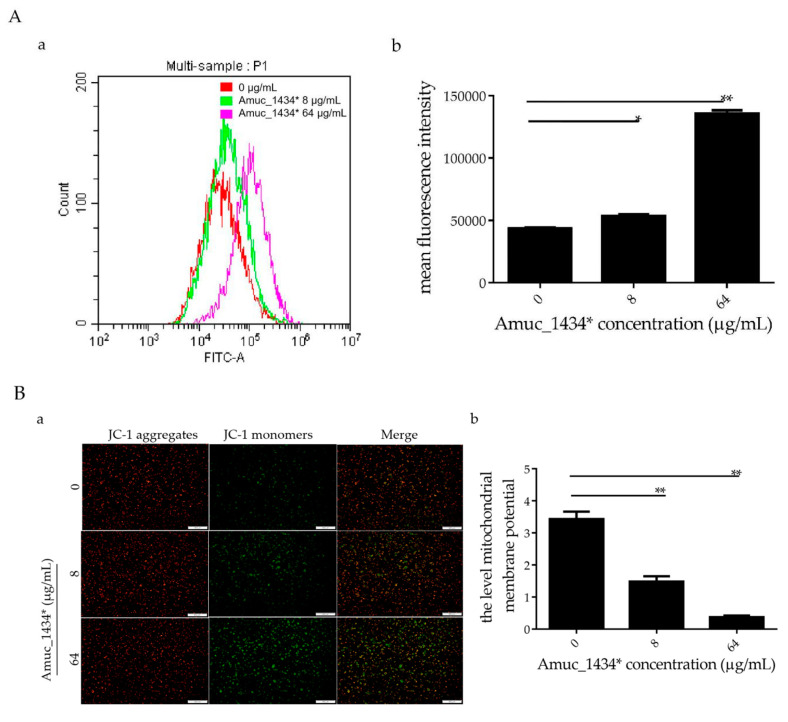
Amuc_1434* promoted a change of cellular redox status and mitochondrial dysfunction in LS174T cells. LS174T cells were treated with the indicated concentrations of Amuc_1434* for 24 h. (**A**) Reactive oxygen species (ROS) production was examined by flow cytometric analysis (**a**) with 2,7-Dichlorodi-hydrofluorescein diacetate (DCFH-DA) staining, (**b**) mean fluorescence intensity of ROS for DCFH-DA. Data are expressed as the mean ± standard deviation of three separate experiments, *: *p* < 0.05; **: *p* < 0.01. (**B**) Mitochondrial membrane potential (Δ ψ) was detected by fluorescence microscopy using JC-1 staining. (**a**) Fluorescent micrographs of mitochondrial membrane potential obtained in Amuc_1434*-treated LS174T cells at 24 h, bar = 500 μm; (**b**) Quantification of relative fluorescent intensity per cell determined by Image J software. Values represent as ratio of red to green fluorescence (**: *p* < 0.01, *n* = 3). Data are mean ± S.E.M.

**Figure 5 ijms-21-03385-f005:**
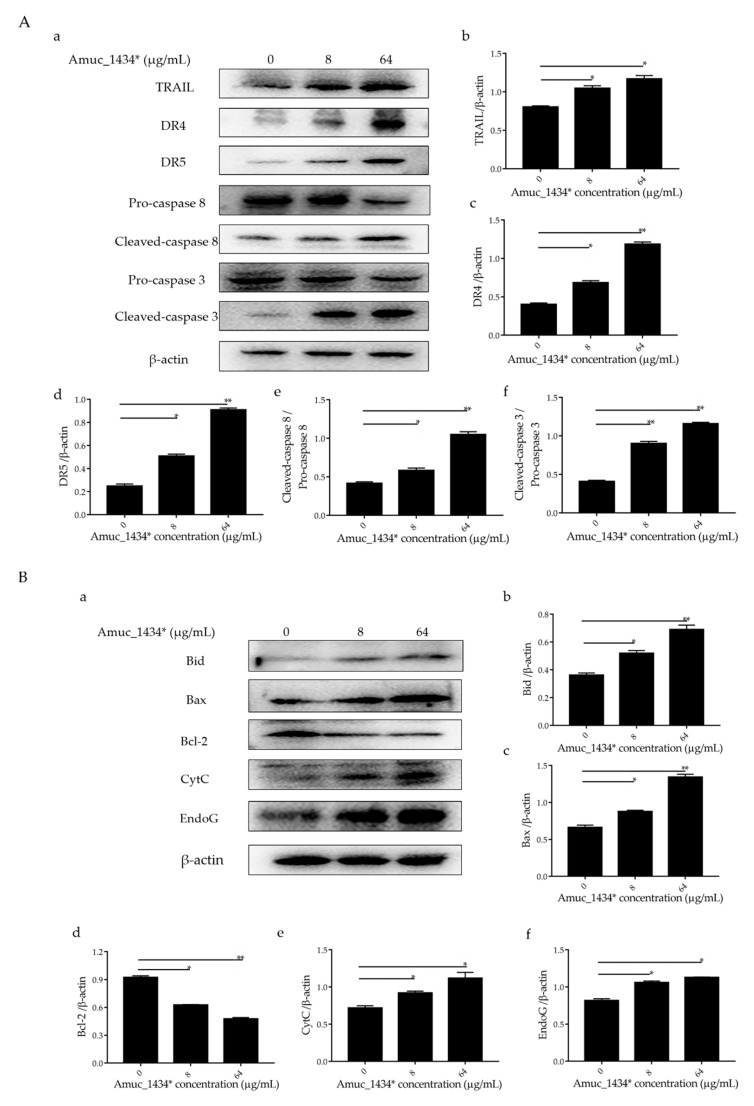
Amuc_1434* mediated the activation of the apoptosis pathway in LS174T cells. (**A**) The expression of death receptor 4 (DR4), death receptor 5 (DR5), cysteinyl aspartate specific proteinase 8 (caspase 8) and cysteinyl aspartate specific proteinase 3 (caspase 3) induced by Amuc_1434* in LS174T cells was dependent on tumor necrosis factor-related apoptosis-inducing ligand (TRAIL). (**a**) LS174T cells were treated with 8 and 64 μg/mL Amuc_1434* for 24 h, respectively. The cell lysates were analyzed by Western blot. (**b**–**d**) The quantification of TRAIL, DR4, and DR5 levels relative to that of the internal control (β-actin). (**e**) The cleaved-caspase 8/pro-caspase 8 ratio and (**f**) the cleaved-caspase 3/pro-caspase 3 ratio. (Data are expressed as mean ± standard deviation, *n* = 3, *: *p* < 0.05; **: *p* < 0.01.) (**B**) Amuc_1434* indirectly affected the expression of proteins in the mitochondrial apoptosis pathway. (**a**) B-cell lymphoma-2 interacting-domain death agonist (Bid), B-cell lymphoma-2-Associated X Protein (Bax), B-cell lymphoma-2 (Bcl-2), Cytochrome c (CytC), and Endonuclease G (EndoG) proteins of LS174T cells were detected using Western blot analysis after treatment with Amuc_1434* for 24 h. β-actin was used as the loading control. (**b**–**f**) The results of the Western blot were analyzed using Image J software. (Data are expressed as mean ± standard deviation, *n* = 3, *: *p* < 0.05; **: *p* < 0.01.).

**Figure 6 ijms-21-03385-f006:**
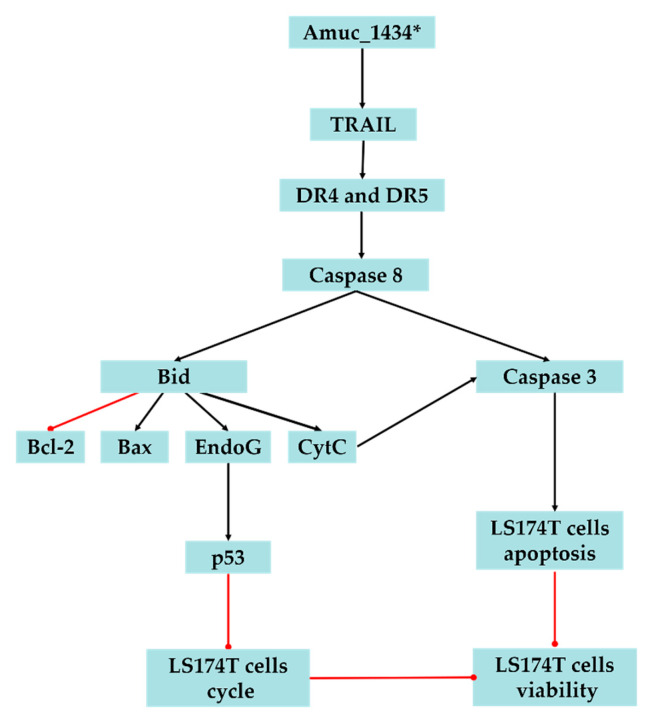
Proposed model of Amuc_1434* effects on LS174T cells. The black lines represent facilitation and the red lines represent inhibition.

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
