# Peer review of "Akkermansia muciniphila Aspartic Protease Amuc_1434* Inhibits Human Colorectal Cancer LS174T Cell Viability via TRAIL-Mediated Apoptosis Pathway"

_ijms, 2020, doi:10.3390/ijms21093385_

Round 1
Reviewer 1 Report
This paper is related to the induction of apoptosis of Amuc_1343 * for colon cancer.
The authors reported that Amuc_1434 * induces apoptosis of colorectal cancer cells through TRAIL signaling.
The results presented in this paper demonstrate well the hypotheses presented by the authors.
Also, the paper is well written. However, some modifications are required for this article to be published in this journal.
1. The separation process for Amuc_1434 * should be simplified.
2. In order to prove the induction of apoptosis through TRAIL signaling of Amuc_1434 *, it is necessary to investigate the effect of Amuc_1434 * on apoptosis according to TRAIL inhibition.
Reviewer 2 Report
Comments for revision:
- Data should be shown that the cell death phenotype is dependent on Muc2 degradation by the protease by inhibiting protease activity or some other method.
- Figure 3B, Please clarify what is being graphed here. It does not appear to be representative of the flow data in A. Or, more representative flow data should be shown in A.
- Figure 3A, axes should be labeled with the respective dyes.
- Figure 4A, overlaying the graphs would be a better way to show the differences.
- Figure 4B, JC-1 staining should be quantified.
- Figure legends or methods should describe the number of technical and biological replicates.
- Discussion should include a mechanism for how Muc2 degradation leads to upregulation of the TRAIL pathway. If you block TRAIL, is cell death blocked?
Reviewer 3 Report
The paper was well prepared but some minor defects were found. Authors ensure using statistical analysis expression of the results is as mean ±
313 standard deviation nut theu presented in graphics the mean and SEM. It should be clarified.
Round 2
Reviewer 2 Report
The authors have addressed my comments.